# An Equivalence between Loss Functions and Non-Uniform Sampling in Experience Replay

**Scott Fujimoto, David Meger, Doina Precup**
Mila, McGill University
scott.fujimoto@mail.mcgill.ca

## Abstract

Prioritized Experience Replay (PER) is a deep reinforcement learning technique in which agents learn from transitions sampled with non-uniform probability proportionate to their temporal-difference error. We show that any loss function evaluated with non-uniformly sampled data can be transformed into another uniformly sampled loss function with the same expected gradient. Surprisingly, we find in some environments PER can be replaced entirely by this new loss function without impact to empirical performance. Furthermore, this relationship suggests a new branch of improvements to PER by correcting its uniformly sampled loss function equivalent. We demonstrate the effectiveness of our proposed modifications to PER and the equivalent loss function in several MuJoCo and Atari environments.

## 1 Introduction

The use of non-uniform sampling in deep reinforcement learning originates from a technique known as prioritized experience replay (PER) [1], in which high error transitions are sampled with increased probability, enabling faster propagation of rewards and accelerating learning by focusing on the most critical data. An ablation study over the many improvements to deep Q-learning [2] found PER to be the most critical extension for overall performance. However, while the motivation for PER is intuitive, this commonly used technique lacks a critical theoretical foundation. In this paper, we develop analysis enabling us to understand the benefits of non-uniform sampling and suggest modifications to PER that both simplify and improve the empirical performance of the algorithm.

In deep reinforcement learning, value functions are approximated with deep neural networks, trained with a loss over the temporal-difference error of previously seen transitions. The expectation of this loss, over the sampling distribution of transitions, determines the effective gradient which is used to find minima in the network. By biasing the sampling distribution, PER effectively changes the expected gradient. Our main theoretical contribution shows the expected gradient of a loss function minimized on data sampled with non-uniform probability, is equal to the expected gradient of another, distinct, loss function minimized on data sampled uniformly. This relationship can be used to transform any loss function into a prioritized sampling scheme with a new loss function or vice-versa. We can use this transformation to develop a concrete understanding about the benefits of non-uniform sampling methods such as PER, as well as facilitate the design of novel prioritized methods. Our discoveries can be summarized as follows:

**Loss function and prioritization should be tied.** The key implication of this result is that the design of prioritized sampling methods should not be considered in isolation from the loss function. We can use this connection to analyze the correctness of methods which use non-uniform sampling by transforming the loss into the uniform-sampling equivalent and considering whether the new loss is in line with the target objective. In particular, we find the PER objective can be unbiased, even without importance sampling, if the loss function is chosen correctly.

**Variance reduction.** This relationship in expected gradients brings us to a deeper understanding of the benefits of prioritization. We show that the variance of the sampled gradient can be reduced by transforming a uniformly sampled loss function into a carefully chosen prioritization scheme and corresponding loss function. This means that non-uniform sampling is almost always favorable to uniform sampling, at the cost of additional algorithmic and computational complexity.

**Empirically unnecessary.** While non-uniform sampling is theoretically favorable, in many cases the variance reducing properties will be minimal. Perhaps most unexpectedly, we find in a standard benchmark problem, prioritized sampling can be replaced with uniform sampling and a modified loss function, without affecting performance. This result suggests some of the benefit of prioritized experience replay comes from the change in expected gradient, rather than the prioritization itself.

We introduce Loss-Adjusted Prioritized (LAP) experience replay and its uniformly sampled loss equivalent, Prioritized Approximation Loss (PAL). LAP simplifies PER by removing unnecessary importance sampling ratios and setting the minimum priority to be 1, which reduces bias and the likelihood of dead transitions with near-zero sampling probability in a principled manner. On the other hand, our loss function PAL, which resembles a weighted variant of the Huber loss, computes the same expected gradient as LAP, and can be added to any deep reinforcement learning algorithm in only a few lines of code. We evaluate both LAP and PAL on the suite of MuJoCo environments [3] and a set of Atari games [4]. Across both domains, we find both of our methods outperform the vanilla algorithms they modify. In the MuJoCo domain, we find significant gains over the state-of-the-art in the hardest task, Humanoid. All code is open-sourced (`https://github.com/sfujim/LAP-PAL`).

## 2 Related Work

The use of prioritization in reinforcement learning originates from prioritized sweeping for value iteration [5, 6, 7] to increase learning speed, but has also found use in modern applications for importance sampling over trajectories [8] and learning from demonstrations [9, 10]. Prioritized experience replay [1] is one of several popular improvements to the DQN algorithm [11, 12, 13, 14] and has been included in many algorithms combining multiple improvements [15, 2, 16, 17]. Variations of PER have been proposed for considering sequences of transitions [18, 19, 20, 21], or optimizing the prioritization function [22]. Alternate replay buffers have been proposed to favor recent transitions without explicit prioritization [23, 24]. The composition and size of the replay buffer has been studied [25, 26, 27, 28], as well as prioritization in simple environments [29].

Non-uniform sampling has been used in the context of training deep networks faster by favoring informative samples [30], or by distributed and non-uniform sampling where data points are re-weighted by importance sampling ratios [31, 32]. There also exists other importance sampling approaches [33, 34] which re-weight updates to reduce variance. In contrast, our method avoids the use of importance sampling by considering changes to the loss function itself and focuses on the context of deep reinforcement learning.

## 3 Preliminaries

We consider a standard reinforcement learning problem setup, framed by a Markov decision process (MDP), a 5-tuple $(\mathcal{S}, \mathcal{A}, \mathcal{R}, p, \gamma)$, with state space $\mathcal{S}$, action space $\mathcal{A}$, reward function $\mathcal{R}$, dynamics model $p$, and discount factor $\gamma$. The behavior of a reinforcement learning agent is defined by its policy $\pi : \mathcal{S} \to \mathcal{A}$. The performance of a policy $\pi$ can be defined by the value function $Q^\pi$, the expected sum of discounted rewards when following $\pi$ after performing the action $a$ in the state $s$: $Q^\pi(s, a) = \mathbb{E}_\pi[\sum_{t=0}^\infty \gamma^t r_{t+1} | s_0 = s, a_0 = a]$. The value function can be determined from the Bellman equation [35]: $Q^\pi(s, a) = \mathbb{E}_{r, s' \sim p; a' \sim \pi}[r + \gamma Q^\pi(s', a')]$.

In deep reinforcement learning algorithms, such as the Deep Q-Network algorithm (DQN) [36, 11], the value function is approximated by a neural network $Q_\theta$ with parameters $\theta$. Given transitions $i = (s, a, r, s')$, DQN is trained by minimizing a loss $\mathcal{L}(\delta(i))$ on the temporal-difference (TD) error $\delta(i)$ [37], the difference between the network output $Q(i) = Q_\theta(i)$ and learning target $y(i)$:

$$\delta(i) = Q(i) - y(i), \quad y(i) = r + \gamma Q_{\theta'}(s', a'). \tag{1}$$

Transitions $i \in \mathcal{B}$ are sampled from an experience replay buffer $\mathcal{B}$ [38], a data set containing previously experienced transitions. The target $y(i)$ uses a separate target network $Q_{\theta'}$ with parameters $\theta'$, which

are frozen to maintain a fixed target over multiple updates, and then updated to copy the parameters $\theta$ after a set number of learning steps. This loss is averaged over a mini-batch of size $M$: $\frac{1}{M}\sum_i \mathcal{L}(\delta(i))$. For analysis, the size of $M$ is unimportant as it does not affect the expected loss or gradient.

Our analysis revolves around the gradient $\nabla_Q$ of different loss functions with respect to the output of the value network $Q_\theta$, noting $\delta(i)$ is a function of $Q(i)$. In this work, we focus on three loss functions, which we define over the TD error $\delta(i)$ of a transition $i$: the L1 loss $\mathcal{L}_{L1}(\delta(i)) = |\delta(i)|$ with a gradient $\nabla_Q\mathcal{L}_{L1}(\delta(i)) = \text{sign}(\delta(i))$, mean squared error (MSE) $\mathcal{L}_{MSE}(\delta(i)) = 0.5\delta(i)^2$ with a gradient $\nabla_Q\mathcal{L}_{MSE}(\delta(i)) = \delta(i)$, and the Huber loss [39]:

$$\mathcal{L}_{\text{Huber}}(\delta(i)) = \begin{cases} 0.5\delta(i)^2 & \text{if } |\delta(i)| \leq \kappa, \\ \kappa(|\delta(i)| - 0.5\kappa) & \text{otherwise,} \end{cases} \tag{2}$$

where generally $\kappa = 1$, giving an equivalent gradient to MSE or L1 loss, depending on $|\delta(i)|$.

**Prioritized Experience Replay.** Prioritized experience replay (PER) [1] is a non-uniform sampling scheme for replay buffers where transitions are sampled in proportion to their temporal-difference (TD) error. The intuitive argument behind PER is that training on the highest error samples will result in the largest performance gain.

PER makes two changes to the traditional uniformly sampled replay buffer. Firstly, the probability of sampling a transition $i$ is proportionate to the absolute TD error $|\delta(i)|$, set to the power of a hyper-parameter $\alpha$ to smooth out extremes:

$$p(i) = \frac{|\delta(i)|^\alpha + \epsilon}{\sum_j (|\delta(j)|^\alpha + \epsilon)}, \tag{3}$$

where a small constant $\epsilon$ is added to ensure each transition is sampled with non-zero probability. This $\epsilon$ is necessary as often the current TD error is approximated by the TD error when $i$ was last sampled. For our theoretical analysis we will treat $\delta(i)$ as the current TD error and assume $\epsilon = 0$.

Secondly, given PER favors transitions with high error, in a stochastic setting it will shift the distribution of $s'$ in the expectation $\mathbb{E}_{s'}[Q(s', a')]$. This is corrected by weighted importance sampling ratios $w(i)$, which reduce the influence of high priority transitions using a ratio between Equation (3) and the uniform probability $\frac{1}{N}$, where $N$ is the total number of elements contained in the buffer:

$$\mathcal{L}_{\text{PER}}(\delta(i)) = w(i)\mathcal{L}(\delta(i)), \qquad w(i) = \frac{\hat{w}(i)}{\max_j \hat{w}(j)}, \qquad \hat{w}(i) = \left(\frac{1}{N} \cdot \frac{1}{p(i)}\right)^\beta. \tag{4}$$

The hyper-parameter $\beta$ is used to smooth out high variance importance sampling weights. The value of $\beta$ is annealed from an initial value $\beta_0$ to 1, to eliminate any bias from distributional shift.

## 4  The Connection between Sampling and Loss Functions

In this section, we present our general results which show the expected gradient of a loss function with non-uniform sampling is equivalent to the expected gradient of another loss function with uniform sampling. This relationship provides an approach to analyze methods which use non-uniform sampling, by considering whether the uniformly sampled loss with equivalent expected gradient is reasonable, with regards to the learning objective. All proofs are in the supplementary material.

To build intuition, consider that the expected gradient of a generic loss $\mathcal{L}_1$ on the TD error $\delta(i)$ of transitions $i$ sampled by a distribution $\mathcal{D}_1$, can be determined from another distribution $\mathcal{D}_2$ by using the importance sampling ratio $\frac{p_{\mathcal{D}_1}(i)}{p_{\mathcal{D}_2}(i)}$:

$$\underbrace{\mathbb{E}_{i\sim\mathcal{D}_1}[\nabla_Q\mathcal{L}_1(\delta(i))]}_{\text{expected gradient of } \mathcal{L}_1 \text{ under } \mathcal{D}_1} = \mathbb{E}_{i\sim\mathcal{D}_2}\left[\frac{p_{\mathcal{D}_1}(i)}{p_{\mathcal{D}_2}(i)}\nabla_Q\mathcal{L}_1(\delta(i))\right]. \tag{5}$$

Suppose we introduce a second loss $\mathcal{L}_2$, where the gradient of $\mathcal{L}_2$ is the inner expression of the RHS of Equation (5): $\nabla_Q\mathcal{L}_2(\delta(i)) = \frac{p_{\mathcal{D}_1}(i)}{p_{\mathcal{D}_2}(i)}\nabla_Q\mathcal{L}_1(\delta(i))$. Then the expected gradient of $\mathcal{L}_1$ under $\mathcal{D}_1$ and $\mathcal{L}_2$ under $\mathcal{D}_2$ would be equal $\mathbb{E}_{\mathcal{D}_1}[\nabla_Q\mathcal{L}_1(\delta(i))] = \mathbb{E}_{\mathcal{D}_2}[\nabla_Q\mathcal{L}_2(\delta(i))]$. It may first seem unlikely that this relationship would exist in practice. However, defining $\mathcal{D}_1$ to be the uniform distribution $\mathcal{U}$

over a finite data set $\mathcal{B}$, and $\mathcal{D}_2$ to be a prioritized sampling scheme $p(i) = \frac{|\delta(i)|}{\sum_{j \in \mathcal{B}} |\delta(j)|}$, we have the following relationship between MSE and L1:

$$\underbrace{\mathbb{E}_{\mathcal{U}}[\nabla_Q \mathcal{L}_{\text{MSE}}(\delta(i))]}_{\text{expected gradient of MSE under } \mathcal{U}} = \underbrace{\mathbb{E}_{\mathcal{D}_2}\left[\frac{\sum_j \delta(j)}{N|\delta(i)|}\delta(i)\right]}_{\text{by Equation (5)}} \propto \mathbb{E}_{\mathcal{D}_2}\underbrace{[\text{sign}(\delta(i))]}_{\nabla_Q \mathcal{L}_{\text{L1}}(\delta(i))} = \underbrace{\mathbb{E}_{\mathcal{D}_2}[\nabla_Q \mathcal{L}_{\text{L1}}(\delta(i))]}_{\text{expected gradient of L1 under } \mathcal{D}_2}$$

(6)

This means the L1 loss, sampled non-uniformly, has the same expected gradient direction as MSE, sampled uniformly. We emphasize that the distribution $\mathcal{D}$ is very similar to the priority scheme in PER. In the following sections we will generalize this relationship, discuss the benefits to prioritization, and derive a uniformly sampled loss function with the same expected gradient as PER.

We describe our main result in Theorem 1 which formally describes the relationship between a loss $\mathcal{L}_1$ on uniformly sampled data, and a loss $\mathcal{L}_2$ on data sampled according to some priority scheme $pr$, such that $p(i) = \frac{pr(i)}{\sum_j pr(j)}$. As suggested above, this relationship draws a similarity to importance sampling, where the ratio between distributions is absorbed into the loss function.

**Theorem 1** *Given a data set $\mathcal{B}$ of $N$ items, loss functions $\mathcal{L}_1$ and $\mathcal{L}_2$, and priority scheme $pr$, the expected gradient of $\mathcal{L}_1(\delta(i))$, where $i \in \mathcal{B}$ is sampled uniformly, is equal to the expected gradient of $\mathcal{L}_2(\delta(i))$, where $i$ is sampled with priority $pr$, if $\nabla_Q \mathcal{L}_1(\delta(i)) = \frac{1}{\lambda}pr(i)\nabla_Q \mathcal{L}_2(\delta(i))$ for all $i$, where $\lambda = \frac{\sum_j pr(j)}{N}$.*

Theorem 1 describes the conditions for a uniformly sampled loss and non-uniformly sampled loss to have the same expected gradient. This additionally provides a recipe for transforming a given loss function $\mathcal{L}_2$ with non-uniform sampling into an equivalent loss function $\mathcal{L}_1$ with uniform sampling.

**Corollary 1** *Theorem 1 is satisfied by any two loss functions $\mathcal{L}_1$, where $i \in \mathcal{B}$ is sampled uniformly, and $\mathcal{L}_2$, where $i$ is sampled with respect to priority $pr$, if $\mathcal{L}_1(\delta(i)) = \frac{1}{\lambda}|pr(i)|_\times \mathcal{L}_2(\delta(i))$ for all $i$, where $\lambda = \frac{\sum_j pr(j)}{N}$ and $|\cdot|_\times$ is the stop-gradient operation.*

As noted by our example in the beginning of the section, often the loss function equivalent of a prioritization method is surprisingly simple. The converse relationship can also be determined. By carefully choosing how the data is sampled, a loss function $\mathcal{L}_1$ can be transformed into (almost) any other loss function $\mathcal{L}_2$ in expectation.

**Corollary 2** *Theorem 1 is satisfied by any two loss functions $\mathcal{L}_1$, where $i \in \mathcal{B}$ is sampled uniformly, and $\lambda\mathcal{L}_2$, where $i$ is sampled with respect to priority $pr$ and $\lambda = \frac{\sum_j pr(j)}{N}$, if $\text{sign}(\nabla_Q \mathcal{L}_1(\delta(i))) = \text{sign}(\nabla_Q \mathcal{L}_2(\delta(i)))$ and $pr(i) = \frac{\nabla_Q \mathcal{L}_1(\delta(i))}{\nabla_Q \mathcal{L}_2(\delta(i))}$ for all $i$.*

Note that $\text{sign}(\nabla_Q \mathcal{L}_1(\delta(i))) = \text{sign}(\nabla_Q \mathcal{L}_2(\delta(i)))$ is only required as $pr(i)$ must be non-negative, and is trivially satisfied by all loss functions which aim to minimize the distance between the output $Q$ and a given target. In this instance, the non-uniform sampling acts similarly to an importance sampling ratio, re-weighting the gradient of $\mathcal{L}_2$ to match the gradient of $\mathcal{L}_1$ in expectation. Corollary 2 is perhaps most interesting when $\mathcal{L}_2$ is the L1 loss as $\nabla_Q \mathcal{L}_{\text{L1}}(\delta(i)) = \pm 1$, setting $pr(i) = |\nabla_Q \mathcal{L}_1(\delta(i))|$ allows us to transform the loss $\mathcal{L}_1$ into a prioritization scheme. It turns out that transforming a given loss function into its prioritized variant with a L1 loss can be used to reduce the variance of the gradient.

**Observation 1** *Given a data set $\mathcal{B}$ of $N$ items and loss function $\mathcal{L}_1$, the gradient of the loss function $\lambda\mathcal{L}_{\text{L1}}(\delta(i))$, where $i \in \mathcal{B}$ is sampled with priority $pr(i) = |\nabla_Q \mathcal{L}_1(\delta(i))|$ and $\lambda = \frac{\sum_j pr(j)}{N}$, will have lower (or equal) variance than the gradient of $\mathcal{L}_1(\delta(i))$, where $i$ is sampled uniformly.*

In fact, we can generalize this observation one step further, and show that the L1 loss, and corresponding priority scheme, produce the lowest variance of all possible loss functions with the same expected gradient.

**Theorem 2** *Given a data set $\mathcal{B}$ of $N$ items and loss function $\mathcal{L}_1$, consider the loss function $\lambda\mathcal{L}_2(\delta(i))$, where $i \in \mathcal{B}$ is sampled with priority $pr$ and $\lambda = \frac{\sum_j pr(j)}{N}$, such that Theorem 1 is satisfied. The variance of $\nabla_Q \lambda\mathcal{L}_2(\delta(i))$ is minimized when $\mathcal{L}_2 = \mathcal{L}_{\text{L1}}$ and $pr(i) = |\nabla_Q \mathcal{L}_1(\delta(i))|$.*

Intuitively, Theorem 2 applies by taking equally-sized gradient steps, rather than intermixing small and large steps. These results suggest a simple recipe for reducing the variance of any loss function while keeping the expected gradient unchanged, by using the L1 loss and corresponding prioritization.

## 5 Corrections to Prioritized Experience Replay

We now consider prioritized experience replay (PER) [1] and by following Theorem 1 from the previous section, derive its uniformly sampled equivalent loss function. This relationship allows us to consider corrections and possible simplifications. All proofs are in the supplementary material.

### 5.1 An Equivalent Loss Function to Prioritized Experience Replay

We begin by deriving the uniformly sampled equivalent loss function to PER. Our general finding is that when mean squared error (MSE) is used, including a subset of cases in the Huber loss [39], PER optimizes a loss function on the TD error to a power higher than two. This means PER may favor outliers in its estimate of the expectation in the temporal difference target, rather than learn the mean. Furthermore, we find that the importance sampling ratios used by PER can be absorbed into the loss function themselves, which gives an opportunity to simplify the algorithm.

Firstly, from Theorem 1, we can derive a general result on a loss function $\frac{1}{\tau}|\delta(i)|^\tau$ when used in conjunction with PER.

**Theorem 3** *The expected gradient of a loss $\frac{1}{\tau}|\delta(i)|^\tau$, where $\tau > 0$, when used with PER is equal to the expected gradient of the following loss when using a uniformly sampled replay buffer:*

$$\mathcal{L}_{\text{PER}}^\tau(\delta(i)) = \frac{\eta N}{\tau + \alpha - \alpha\beta}|\delta(i)|^{\tau + \alpha - \alpha\beta}, \qquad \eta = \frac{\min_j |\delta(j)|^{\alpha\beta}}{\sum_j |\delta(j)|^\alpha}. \tag{7}$$

Noting that DQN [11] traditionally uses the Huber loss, we can now show the form of the uniformly sampled loss function with the same expected gradient as PER, when used with DQN.

**Corollary 3** *The expected gradient of the Huber loss when used with PER is equal to the expected gradient of the following loss when using a uniformly sampled replay buffer:*

$$\mathcal{L}_{\text{PER}}^{\text{Huber}}(\delta(i)) = \frac{\eta N}{\tau + \alpha - \alpha\beta}|\delta(i)|^{\tau + \alpha - \alpha\beta}, \qquad \tau = \begin{cases} 2 & \text{if } |\delta(i)| \leq 1, \\ 1 & \text{otherwise,} \end{cases} \qquad \eta = \frac{\min_j |\delta(j)|^{\alpha\beta}}{\sum_j |\delta(j)|^\alpha}. \tag{8}$$

To understand the significance of Corollary 3 and what it says about the objective in PER, first consider the following two observations on MSE and L1:

**Observation 2** *(MSE) Let $\mathcal{B}(s,a) \subset \mathcal{B}$ be the subset of transitions containing $(s,a)$ and $\delta(i) = Q(i) - y(i)$. If $\nabla_Q \mathbb{E}_{i \sim \mathcal{B}(s,a)}[0.5\delta(i)^2] = 0$ then $Q(s,a) = \text{mean}_{i \in \mathcal{B}(s,a)} y(i)$.*

**Observation 3** *(L1 Loss) Let $\mathcal{B}(s,a) \subset \mathcal{B}$ be the subset of transitions containing $(s,a)$ and $\delta(i) = Q(i) - y(i)$. If $\nabla_Q \mathbb{E}_{i \sim \mathcal{B}}[|\delta(i)|] = 0$ then $Q(s,a) = \text{median}_{i \in \mathcal{B}(s,a)} y(i)$.*

From Corollary 3 and the aforementioned observations, we can make several statements:

**The PER objective is biased if $\tau + \alpha - \alpha\beta \neq 2$.** The implication of Observation 2 is that minimizing MSE gives us an estimate of the target of interest, the expected temporal-difference target $y(i) = r + \gamma \mathbb{E}_{s',a'}[Q(s',a')]$. It follows that optimizing a loss $|\delta(i)|^\tau$ with PER, such that $\tau + \alpha - \alpha\beta \neq 2$, produces a biased estimate of the target. On the other hand, we argue that not all bias is equal. From Observation 3 we see that minimizing a L1 loss gives the median target, rather than the expectation. Given the effects of function approximation and bootstrapping in deep reinforcement learning, one could argue the median is a reasonable loss function, due to its robustness properties. A likely possibility is that an in-between of MSE and L1 would provide a balance of robustness and "correctness". Looking at Equation (7), it turns out this is what PER does when combined with an L1 loss, as the loss will be to a power of $1 + \alpha - \alpha\beta \in [1, 2]$ for $\alpha \in (0, 1]$ and $\beta \in [0, 1]$. However, when combined with MSE, if $\beta < 1$, then $2 + \alpha - \alpha\beta > 2$ for $\alpha \in (0, 1]$. This means while MSE is

normally minimized by the mean, when combined with PER, the loss will be minimized by some expression which may favor outliers. This bias explains the poor performance of PER with standard algorithms in continuous control which rely on MSE [23, 22, 40, 24].

**Importance sampling can be avoided.** PER uses importance sampling (IS) to re-weight the loss function as an approach to reduce the bias introduced by prioritization. We note that PER with MSE is unbiased if the IS hyper-parameter $\beta = 1$. However, our theoretical results show the prioritization is absorbed into the expected gradient. Consequently, PER can be "unbiased" even without using IS ($\beta = 0$) if the expected gradient is still meaningful. As discussed above, this simply means selecting $\alpha$ and $\tau$ such that $\tau + \alpha \leq 2$. Furthermore, this allows us to avoid any bias from weighted IS.

## 5.2 Loss-Adjusted Prioritized Experience Replay

We now utilize some of the understanding developed in the previous section for the design of a new prioritized experience replay scheme. We call our variant Loss-Adjusted Prioritized (LAP) experience replay, as our theoretical results argue that the choice of loss function should be closely tied to the choice of prioritization. Mirroring LAP, we also introduce a uniformly sampled loss function, Prioritized Approximation Loss (PAL) with the same expected gradient as LAP. When the variance reduction from prioritization is minimal, PAL can be used as a simple and computationally efficient replacement for LAP, requiring only an adjustment to the loss function used to train the Q-network.

With these ideas in mind, we can now introduce our corrections to the traditional prioritized experience replay, which we combine into a simple prioritization scheme which we call the Loss-Adjusted Prioritized (LAP) experience replay. The simplest variant of LAP would use the L1 loss, as suggested by Theorem 2, and sample values with priority $pr(i) = |\delta(i)|^{\alpha}$. Following Theorem 1, we can see the expected gradient of this approach is proportional to $|\delta(i)|^{1+\alpha}$, when sampled uniformly.

However, in practice, L1 loss may not be preferable as each update takes a constant-sized step, possibly overstepping the target if the learning rate is too large. Instead, we apply the commonly used Huber loss, with $\kappa = 1$, which swaps from L1 to MSE when the error falls below a threshold of 1, scaling the appropriately gradient as $\delta(i)$ approaches 0. When $|\delta(i)| < 1$ and MSE is applied, if we want to avoid the bias introduced from using MSE and prioritization, samples with error below 1 should be sampled uniformly. This can be achieved with a priority scheme $pr(i) = \max(|\delta(i)|^{\alpha}, 1)$, where samples with otherwise low priority are clipped to be at least 1. We denote this algorithm LAP and it can be described by non-uniform sampling and the Huber loss:

$$p(i) = \frac{\max(|\delta(i)|^{\alpha}, 1)}{\sum_j \max(|\delta(j)|^{\alpha}, 1)}, \qquad \mathcal{L}_{\text{Huber}}(\delta(i)) = \begin{cases} 0.5\delta(i)^2 & \text{if } |\delta(i)| \leq 1, \\ |\delta(i)| & \text{otherwise.} \end{cases} \tag{9}$$

On top of correcting the outlier bias, this clipping reduces the likelihood of dead transitions that occur when $p(i) \approx 0$, eliminating the need for a $\epsilon$ hyper-parameter, which is added to the priority of each transition in PER. LAP maintains the variance reduction properties defined by our theoretical analysis as it uses the L1 loss on all samples with large errors. Following our ideas from the previous section, we can transform LAP into its mirrored loss function with an equivalent expected gradient, which we denote Prioritized Approximation Loss (PAL). PAL can be derived following Corollary 1:

$$\mathcal{L}_{\text{PAL}}(\delta(i)) = \frac{1}{\lambda} \begin{cases} 0.5\delta(i)^2 & \text{if } |\delta(i)| \leq 1, \\ \frac{|\delta(i)|^{1+\alpha}}{1+\alpha} & \text{otherwise,} \end{cases} \qquad \lambda = \frac{\sum_j \max(|\delta(j)|^{\alpha}, 1)}{N}. \tag{10}$$

**Observation 4** *LAP and PAL have the same expected gradient.*

As PAL and LAP share the same expected gradient, we have a mechanism for analysing both methods. Importantly, we note that loss defined by PAL is never to a power greater than two, meaning the outlier bias from PER has been eliminated. In some cases, we will find PAL is useful as a loss function on its own, and in domains where the variance reducing property from prioritization is unimportant, we should expect their performance to be similar.

## 6 Experiments

We evaluate the benefits of LAP and PAL on the standard suite of MuJoCo [3] continuous control tasks as well as a subset of Atari games, both interfaced through OpenAI gym [41]. For continuous

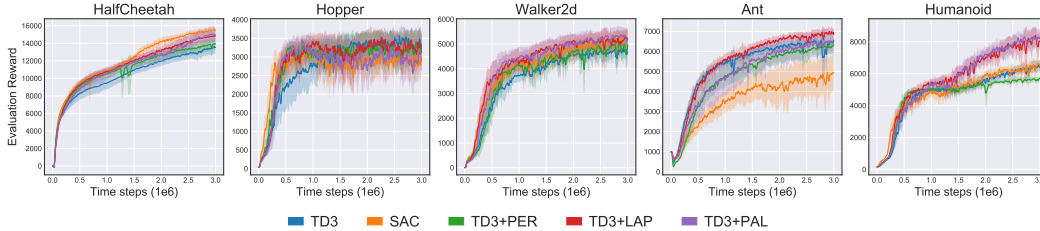

Figure 1: Learning curves for the suite of OpenAI gym continuous control tasks in MuJoCo. Curves are averaged over 10 trials, where the shaded area represents a 95% confidence interval over the trials.

Table 1: Average performance over the last 10 evaluations and 10 trials. $\pm$ captures a 95% confidence interval. Scores are bold if the confidence interval intersects with the confidence interval of the highest performance, except for Hopper and Walker2d where all scores satisfy this condition.

|  | TD3 | SAC | TD3 + PER | TD3 + LAP | TD3 + PAL |
|---|---|---|---|---|---|
| HalfCheetah | $13570.9 \pm 794.2$ | $\mathbf{15511.6 \pm 305.2}$ | $13927.8 \pm 683.9$ | $\mathbf{14836.5 \pm 532.2}$ | $\mathbf{15012.2 \pm 885.4}$ |
| Hopper | $3393.2 \pm 381.9$ | $2851.6 \pm 417.4$ | $3275.5 \pm 451.8$ | $3246.9 \pm 463.4$ | $3129.1 \pm 473.5$ |
| Walker2d | $4692.4 \pm 423.6$ | $5234.4 \pm 346.1$ | $4719.1 \pm 492.0$ | $5230.5 \pm 368.2$ | $5218.7 \pm 422.6$ |
| Ant | $6469.9 \pm 200.3$ | $4923.6 \pm 882.3$ | $6278.7 \pm 311.3$ | $\mathbf{6912.6 \pm 234.4}$ | $\mathbf{6476.2 \pm 640.2}$ |
| Humanoid | $6437.5 \pm 349.3$ | $6580.9 \pm 296.6$ | $5629.3 \pm 174.4$ | $\mathbf{7855.6 \pm 705.9}$ | $\mathbf{8265.9 \pm 519.0}$ |

control, we combine our methods with a state-of-the-art algorithm TD3 [42], which we benchmark against, as well as SAC [43], which have both been shown to outperform other standard algorithms on the MuJoCo benchmark. For Atari, we apply LAP and PAL to Double DQN (DDQN) [12], and benchmark the performance against PER + DDQN. A complete list of hyper-parameters and experimental details are provided in the supplementary material. MuJoCo results are presented in Figure 1 and Table 1 and Atari results are presented in Figure 2 and Table 2.

We find that the addition of either LAP or PAL matches or outperforms the vanilla version of TD3 in all tasks. In the challenging Humanoid task, we find that both LAP and PAL offer a large improvement over the previous state-of-the-art. Interestingly enough, we find no meaningful difference in the performance of LAP and PAL across all tasks. This means that prioritization has little benefit and the improvement comes from the change in expected gradient from using our methods over MSE. Consequently, for MuJoCo environments, non-uniform sampling can be replaced by adjusting the loss function instead. Additionally, we confirm previous results which found PER provides no benefit when added to TD3 [40]. Given prioritization appears to have little impact in this domain, this result is consistent with our theoretical analysis which shows using that MSE with PER introduces bias.

In the Atari domain, we find LAP improves over PER in 9 out of 10 environments, with a mean performance gain of 53%. On the other hand, while PAL offers non-trivial gains over vanilla DDQN, it under-performs PER on 6 of the 10 environments tested. This suggests that prioritization plays a more significant role in the Atari domain, but some of the improvement can still be attributed to the change in expected gradient. As the Atari domain includes games which require longer horizons or sparse rewards, the benefit of prioritization is not surprising.

When $\alpha = 0$, PAL equals a Huber loss with a scaling factor $\frac{1}{\lambda}$. We perform an ablation study to disentangle the importance of the components to PAL. We compare the performance of PAL and the Huber loss with and without $\frac{1}{\lambda}$. The results are presented in the supplementary material. We find the complete loss of PAL achieves the highest performance, and the change in expected gradient from using $\alpha \neq 0$ to be the largest contributing factor.

## 7 Discussion

**Performance.** The performance of PAL is of particular interest due to the simplicity of the method, requiring only a small change to the loss function. In the MuJoCo domain, TD3 + PAL matches the performance of our prioritization scheme while outperforming vanilla TD3 and SAC. Our ablation study shows the performance gain largely comes from the change in expected gradient. We believe the benefit of PAL over MSE is the additional robustness, and the benefit over the Huber loss is a

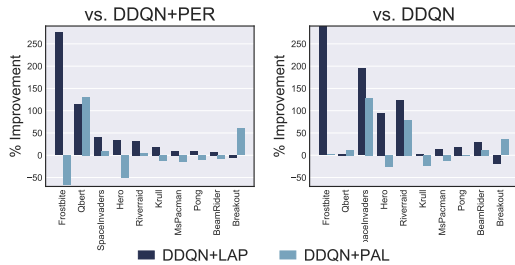

Figure 2: We display the percentage improvement of final scores achieved by DDQN + PAL and DDQN + LAP when compared to DDQN + PER (left) and DDQN (right). Some extreme values are visually clipped.

Table 2: Mean and median percentage improvement of final scores achieved over DDQN and DDQN + PER across 10 Atari games.

|  | Mean % Gain | Median % Gain |
|---|---|---|
| vs. DDQN + PER | | |
| DDQN | -8.06% | -13.00% |
| DDQN + LAP | **+53.38%** | **+24.98%** |
| DDQN + PAL | +4.50% | -8.96% |
| vs. DDQN | | |
| DDQN + PER | +37.65% | +15.24% |
| DDQN + LAP | **+148.16%** | **+24.35%** |
| DDQN + PAL | +20.46% | +6.79% |

better approximation of the mean. There is also a small gain in performance due to the re-scaling used to match the exact expected gradient of LAP. When examining PAL in isolation, it is somewhat unclear why this re-scaling should improve performance. One hypothesis is that as $\frac{1}{\lambda}$ decreases the average error increases, this can be thought of an approach to balance out large gradient steps. On the other hand, the performance gain from LAP is more intuitive to understand, as we correct underlying issues with PER by understanding its relationship with the loss function.

**Alternate Priority schemes.** Our theoretical results define a relationship between the expected gradient of a non-uniformly sampled loss function and a uniformly sampled loss function. This allows us define the optimal variance-reducing prioritization scheme and corresponding loss function. However, there is still more research to do in non-uniform sampling, as often the a uniform sample of the replay buffer is not representative of the data of interest [44], and alternate weightings may be preferred such as the stationary distribution [45, 46, 47] or importance sampling ratios [8]. We hope our results provide a valuable tool for analyzing and understanding alternate prioritization schemes.

**Reproducibility and algorithmic credit assignment.** Our work emphasizes the susceptible nature of deep reinforcement learning algorithms to small changes [48] and the reproducibility crisis [49], as we are able to show significant improvements to the performance of a well-known algorithm with only minor changes to the loss function. This suggests that papers which use intensive hyper-parameter optimization or introduce algorithmic changes without ablation studies may be improving over the original algorithm due to unintended consequences, rather than the proposed method.

# 8 Conclusion

In this paper, we aim to build the theoretical foundations for a commonly used deep reinforcement learning technique known as prioritized experience replay (PER) [1]. To do so, we first show an interesting connection between non-uniform sampling and loss functions. Namely, any loss function can be approximated, in the sense of having the same expected gradient, by a new loss function with some non-uniform sampling scheme, and vice-versa. We use this relationship to show that the prioritized, non-uniformly sampled variant of the loss function has lower variance than the uniformly sampled equivalent. This result suggests that by carefully considering the loss function, using prioritization should outperform the standard setup of uniform sampling.

However, without considering the loss function, prioritization changes the expected gradient. This allows us to develop a concrete understanding of why PER has been shown to perform poorly when combined with continuous control methods in the past [40, 24], due to a bias towards outliers when used with MSE. We introduce a corrected version of PER which considers the loss function, known as Loss-Adjusted Prioritized (LAP) experience replay and its mirrored uniformly sampled loss function equivalent, Prioritized Approximation Loss (PAL). We test both LAP and PAL on standard deep reinforcement learning benchmarks in MuJoCo and Atari, and show their addition improves upon the vanilla algorithm across both domains.

## Broader Impact

Our research focuses on developing the theoretical foundations for a commonly used technique in deep reinforcement learning, prioritized experience replay. Our insights could be used to improve reinforcement learning systems over a wide range of applications such as clinical trial design [50], educational games [51] and recommender systems [52, 53]. Non-uniform sampling may also have benefits for scaling offline reinforcement learning, in particular, when learning from large data sets where sampling only relevant or important data is critical. We expect our impact to be more significant for the reinforcement learning community itself. In the supplementary material we demonstrate our publicly released implementation of PER runs significantly faster (5-17×) than previous implementations published by corporate research groups [54, 55]. This improves the accessibility of non-uniform sampling strategy and state-of-the-art deep reinforcement learning research for groups with resource limitations.

## Acknowledgments and Disclosure of Funding

Scott Fujimoto is supported by a NSERC scholarship as well as the Borealis AI Global Fellowship Award. This research was enabled in part by support provided by Calcul Québec and Compute Canada. We would like to thank Edward Smith for helpful discussions and feedback.

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
