[Supplementary Material]

# A  Detailed Proofs

## A.1  Theorem 1

**Theorem 1** *Given a data set $\mathcal{B}$ of $N$ items, loss functions $\mathcal{L}_1$ and $\mathcal{L}_2$, and priority scheme $pr$, the expected gradient of $\mathcal{L}_1(\delta(i))$, where $i \in \mathcal{B}$ is sampled uniformly, is equal to the expected gradient of $\mathcal{L}_2(\delta(i))$, where $i$ is sampled with priority $pr$, if $\nabla_Q \mathcal{L}_1(\delta(i)) = \frac{1}{\lambda} pr(i) \nabla_Q \mathcal{L}_2(\delta(i))$ for all $i$, where $\lambda = \frac{\sum_j pr(j)}{N}$.*

*Proof.*

$$
\begin{aligned}
\mathbb{E}_{i \sim \mathcal{B}} \left[ \nabla_Q \mathcal{L}_1(\delta(i)) \right] &= \frac{1}{N} \sum_i \nabla_Q \mathcal{L}_1(\delta(i)) \\
&= \frac{1}{N} \sum_i \frac{N}{\sum_j pr(j)} pr(i) \nabla_Q \mathcal{L}_2(\delta(i)) \\
&= \sum_i \frac{pr(i)}{\sum_j pr(j)} \nabla_Q \mathcal{L}_2(\delta(i)) \\
&= \mathbb{E}_{i \sim pr} \left[ \nabla_Q \mathcal{L}_2(\delta(i)) \right].
\end{aligned}
\tag{1}
$$

∎

**Corollary 1** *Theorem 1 is satisfied by any two loss functions $\mathcal{L}_1$, where $i \in \mathcal{B}$ is sampled uniformly, and $\mathcal{L}_2$, where $i$ is sampled with respect to priority $pr$, if $\mathcal{L}_1(\delta(i)) = \frac{1}{\lambda} |pr(i)|_\times \mathcal{L}_2(\delta(i))$ for all $i$, where $\lambda = \frac{\sum_j pr(j)}{N}$ and $| \cdot |_\times$ is the stop-gradient operation.*

*Proof.*

$$
\begin{aligned}
\nabla_Q \mathcal{L}_1(\delta(i)) &= \nabla_Q \frac{1}{\lambda} |pr(i)|_\times \mathcal{L}_2(\delta(i)) \\
&= \frac{1}{\lambda} pr(i) \nabla_Q \mathcal{L}_2(\delta(i)).
\end{aligned}
\tag{2}
$$

Then $\mathcal{L}_1$ and $\mathcal{L}_2$ have the same expected gradient by Theorem 1.

∎

**Corollary 2** *Theorem 1 is satisfied by any two loss functions $\mathcal{L}_1$, where $i \in \mathcal{B}$ is sampled uniformly, and $\lambda \mathcal{L}_2$, where $i$ is sampled with respect to priority $pr$ and $\lambda = \frac{\sum_j pr(j)}{N}$, if $\mathrm{sign}(\nabla_Q \mathcal{L}_1(\delta(i))) = \mathrm{sign}(\nabla_Q \mathcal{L}_2(\delta(i)))$ and $pr(i) = \frac{\nabla_Q \mathcal{L}_1(\delta(i))}{\nabla_Q \mathcal{L}_2(\delta(i))}$ for all $i$.*

*Proof.* Given $\mathrm{sign}(\nabla_Q \mathcal{L}_1(\delta(i))) = \mathrm{sign}(\nabla_Q \mathcal{L}_2(\delta(i)))$, we have $\mathrm{sign}(pr(i)) = 1$, as we cannot sample with negative priority. Theorem 1 is satisfied as:

$$
\begin{aligned}
\frac{1}{\lambda} pr(i) \nabla_Q \lambda \mathcal{L}_2(\delta(i)) &= \frac{\lambda}{\lambda} \cdot \frac{\nabla_Q \mathcal{L}_1(\delta(i))}{\nabla_Q \mathcal{L}_2(\delta(i))} \nabla_Q \mathcal{L}_2(\delta(i)) \\
&= \nabla_Q \mathcal{L}_1(\delta(i)).
\end{aligned}
\tag{3}
$$

∎

## A.2  Theorem 2

**Theorem 2** *Given a data set $\mathcal{B}$ of $N$ items and loss function $\mathcal{L}_1$, consider the loss function $\lambda \mathcal{L}_2(\delta(i))$, where $i \in \mathcal{B}$ is sampled with priority $pr$ and $\lambda = \frac{\sum_j pr(j)}{N}$, such that Theorem 1 is satisfied. The variance of $\nabla_Q \lambda \mathcal{L}_2(\delta(i))$ is minimized when $\mathcal{L}_2 = \mathcal{L}_{\mathrm{L1}}$ and $pr(i) = |\nabla_Q \mathcal{L}_1(\delta(i))|$.*

*Proof.*

Consider the variance of the gradient with prioritized sampling. Note $\text{Var}(x) = \mathbb{E}[x^2] - \mathbb{E}[x]^2$.

$$\text{Var}\left(\nabla_Q \lambda \mathcal{L}_2(\delta(i))\right) = \mathbb{E}_{i \sim pr}\left[\left(\nabla_Q \lambda \mathcal{L}_2(\delta(i))\right)^2\right] - \mathbb{E}_{i \sim pr}\left[\nabla_Q \lambda \mathcal{L}_2(\delta(i))\right]^2$$

$$= \sum_i \frac{pr(i)}{\sum_j pr(j)} \frac{\left(\sum_j pr(j)\right)^2}{N^2} \left(\nabla_Q \mathcal{L}_2(\delta(i))\right)^2 - X \qquad (4)$$

$$= \frac{\sum_j pr(j)}{N^2} \sum_i \nabla_Q \mathcal{L}_1(\delta(i)) \nabla_Q \mathcal{L}_2(\delta(i)) - X,$$

where we define $X = \mathbb{E}_{i \sim \mathcal{B}}\left[\nabla_Q \mathcal{L}_1(\delta(i))\right]^2 = \mathbb{E}_{i \sim pr}\left[\lambda \nabla_Q \mathcal{L}_2(\delta(i))\right]^2$, the square of the unbiased expected gradient.

For L1 loss, noting $\text{sign}(\nabla_Q \mathcal{L}_1(\delta(i))) = \text{sign}(\nabla_Q \mathcal{L}_2(\delta(i)))$, then setting $\mathcal{L}_2 = \mathcal{L}_{\text{L1}}$, we have $pr(i) = |\nabla_Q \mathcal{L}_1(\delta(i))|$ and $\nabla_Q \mathcal{L}_1(\delta(i)) \nabla_Q \mathcal{L}_2(\delta(i)) = |\nabla_Q \mathcal{L}_1(\delta(i))|$, and we can simplify the expression:

$$= \frac{\sum_j |\nabla_Q \mathcal{L}_1(\delta(j))|}{N^2} \sum_i \nabla_Q \mathcal{L}_1(\delta(i)) - X$$

$$= \left(\frac{\sum_j |\nabla_Q \mathcal{L}_1(\delta(j))|}{N}\right)^2 - X. \qquad (5)$$

Now consider a generic prioritization scheme where $\nabla_Q \mathcal{L}_2(\delta(i)) = f(\delta(i))$. To give the same expected gradient, by Theorem 1 we must have $pr(i) = \nabla_Q \mathcal{L}_1(\delta(i))/f(\delta(i))$. To compute the variance, we can insert these terms into Equation (4):

$$= \frac{\sum_j pr(j)}{N^2} \sum_i \nabla_Q \mathcal{L}_1(\delta(i)) \nabla_Q \mathcal{L}_2(\delta(i)) - X$$

$$= \frac{\sum_j \nabla_Q \mathcal{L}_1(\delta(j))/f(\delta(j))}{N^2} \sum_i \nabla_Q \mathcal{L}_1(\delta(i)) f(\delta(i)) - X. \qquad (6)$$

Then choosing $u_j = \frac{\sqrt{\nabla_Q \mathcal{L}_1(\delta(j))/f(\delta(j))}}{\sqrt{N}}$ and $v_j = \frac{\sqrt{\nabla_Q \mathcal{L}_1(\delta(j)) f(\delta(j))}}{\sqrt{N}}$, by Cauchy-Schwarz we have:

$$\left(\frac{\sum_j |\nabla_Q \mathcal{L}_1(\delta(j))|}{N}\right)^2 \leq \frac{\sum_j \nabla_Q \mathcal{L}_1(\delta(j))/f(\delta(j))}{N^2} \sum_i \nabla_Q \mathcal{L}_1(\delta(i)) f(\delta(i)), \qquad (7)$$

with equality if $f(\delta(j)) = \pm c$, where $c$ is a constant.

It follows that the variance is minimized when $\mathcal{L}_2$ is the L1 loss.

∎

**Observation 1** *Given a data set $\mathcal{B}$ of $N$ items and loss function $\mathcal{L}_1$, the gradient of the loss function $\lambda \mathcal{L}_{\text{L1}}(\delta(i))$, where $i \in \mathcal{B}$ is sampled with priority $pr(i) = |\nabla_Q \mathcal{L}_1(\delta(i))|$ and $\lambda = \frac{\sum_j pr(j)}{N}$, will have lower (or equal) variance than the gradient of $\mathcal{L}_1(\delta(i))$, where $i$ is sampled uniformly.*

*Proof.*

This is a direct result of Theorem 2, by noting setting $\mathcal{L}_2 = \mathcal{L}_1$, $pr(i) = \frac{1}{N}$ and $\lambda = \frac{\sum_j pr(j)}{N} = 1$. However, to be comprehensive, consider the variance of $\mathcal{L}_1$ with uniform sampling.

$$\text{Var}\left(\nabla_Q \mathcal{L}_1(\delta(i))\right) = \mathbb{E}_{i \sim \mathcal{B}}\left[\left(\nabla_Q \mathcal{L}_1(\delta(i))\right)^2\right] - \mathbb{E}_{i \sim \mathcal{B}}\left[\nabla_Q \mathcal{L}_1(\delta(i))\right]^2$$

$$= \frac{1}{N} \sum_i \left(\nabla_Q \mathcal{L}_1(\delta(i))\right)^2 - X. \qquad (8)$$

where $X$ is defined as before.

Now by the Cauchy-Schwarz inequality $\left(\sum_j u_j v_j\right)^2 \leq \sum_{j=1}^{N} u_j^2 \sum_{j=1}^{N} v_j^2$ where $u_j = \frac{1}{\sqrt{N}}$ and $v_j = \frac{|\nabla_Q \mathcal{L}_1(\delta(j))|}{\sqrt{N}}$ we have:

$$\left(\frac{\sum_j |\nabla_Q \mathcal{L}_1(\delta(j))|}{N}\right)^2 \leq \frac{1}{N} \sum_i \left(\nabla_Q \mathcal{L}_1(\delta(i))\right)^2, \tag{9}$$

where the LHS is the variance of L1 loss without the $X$ term, Equation (5), and so $\mathrm{Var}\left(\nabla_Q \frac{1}{\lambda} \mathcal{L}_2(\delta(i))\right)$ is less than $\mathrm{Var}\left(\nabla_Q \mathcal{L}_1(\delta(i))\right)$ for all loss functions $\mathcal{L}_1$, when $\mathcal{L}_2 = \mathcal{L}_{\mathrm{L1}}$.

∎

## A.3 Theorem 3

**Theorem 3** *The expected gradient of a loss $\frac{1}{\tau}|\delta(i)|^\tau$, where $\tau > 0$, when used with PER is equal to the expected gradient of the following loss when using a uniformly sampled replay buffer:*

$$\mathcal{L}_{\mathrm{PER}}^\tau(\delta(i)) = \frac{\eta N}{\tau + \alpha - \alpha\beta}|\delta(i)|^{\tau + \alpha - \alpha\beta}, \qquad \eta = \frac{\min_j |\delta(j)|^{\alpha\beta}}{\sum_j |\delta(j)|^\alpha}. \tag{10}$$

*Proof.*

For PER, by definition we have $p(i) = \frac{|\delta(i)|^\alpha}{\sum_{j\in\mathcal{B}} |\delta(j)|^\alpha}$ and $w(i) = \frac{\left(\frac{1}{N} \cdot \frac{1}{p(i)}\right)^\beta}{\max_{j\in\mathcal{B}}\left(\frac{1}{N} \cdot \frac{1}{p(j)}\right)^\beta}$.

Now consider the expected gradient of $\frac{1}{\tau}|\delta(i)|^\tau$, when used with PER:

$$\begin{aligned}
\mathbb{E}_{i\sim\mathrm{PER}}\left[\nabla_Q w(i)\frac{1}{\tau}|\delta(i)|^\tau\right] &= \sum_{i\in\mathcal{B}} w(i)p(i)\nabla_Q \frac{1}{\tau}|\delta(i)|^\tau \\
&= \sum_{i\in\mathcal{B}} \frac{\left(\frac{1}{N} \cdot \frac{1}{p(i)}\right)^\beta}{\max_{j\in\mathcal{B}}\left(\frac{1}{N} \cdot \frac{1}{p(j)}\right)^\beta} \frac{|\delta(i)|^\alpha}{\sum_{j\in\mathcal{B}} |\delta(j)|^\alpha} \mathrm{sign}(\delta(i))|\delta(i)|^{\tau-1} \\
&= \frac{1}{\max_{j\in\mathcal{B}} \frac{1}{|\delta(j)|^{\alpha\beta}} \sum_{j\in\mathcal{B}} |\delta(j)|^\alpha} \sum_{i\in\mathcal{B}} \frac{|\delta(i)|^{\tau+\alpha-1}\mathrm{sign}(\delta(i))}{|\delta(i)|^{\alpha\beta}} \\
&= \eta \sum_{i\in\mathcal{B}} \mathrm{sign}(\delta(i))|\delta(i)|^{\tau+\alpha-\alpha\beta-1}.
\end{aligned} \tag{11}$$

Now consider the expected gradient of $\mathcal{L}_{\mathrm{PER}}^\tau(\delta(i))$:

$$\begin{aligned}
\mathbb{E}_{i\sim\mathcal{B}}\left[\nabla_Q \mathcal{L}_{\mathrm{PER}}^\tau(\delta(i))\right] &= \frac{1}{N}\sum_{i\in\mathcal{B}} \frac{\eta N}{\tau + \alpha - \alpha\beta}\nabla_Q|\delta(i)|^{\tau+\alpha-\alpha\beta} \\
&= \eta \sum_{i\in\mathcal{B}} \mathrm{sign}(\delta(i))|\delta(i)|^{\tau+\alpha-\alpha\beta-1}.
\end{aligned} \tag{12}$$

∎

**Corollary 3** *The expected gradient of the Huber loss when used with PER is equal to the expected gradient of the following loss when using a uniformly sampled replay buffer:*

$$\mathcal{L}_{\mathrm{PER}}^{\mathrm{Huber}}(\delta(i)) = \frac{\eta N}{\tau + \alpha - \alpha\beta}|\delta(i)|^{\tau+\alpha-\alpha\beta}, \qquad \tau = \begin{cases} 2 & \text{if } |\delta(i)| \leq 1, \\ 1 & \text{otherwise,} \end{cases} \qquad \eta = \frac{\min_j |\delta(j)|^{\alpha\beta}}{\sum_j |\delta(j)|^\alpha}. \tag{13}$$

*Proof.* Direct application of Theorem 3 with $\tau = 1$ and $\tau = 2$.

∎

**Observation 2** *(MSE) Let $\mathcal{B}(s,a) \subset \mathcal{B}$ be the subset of transitions containing $(s,a)$ and $\delta(i) = Q(i) - y(i)$. If $\nabla_Q \mathbb{E}_{i \sim \mathcal{B}(s,a)}[0.5|\delta(i)|^2] = 0$ then $Q(s,a) = \text{mean}_{i \in \mathcal{B}(s,a)} y(i)$.*

*Proof.*

$$
\begin{aligned}
& \mathbb{E}_{i \sim \mathcal{B}(s,a)}[\nabla_Q 0.5|\delta(i)|^2] = 0 \\
\Rightarrow\ & \mathbb{E}_{i \sim \mathcal{B}(s,a)}[\delta(i)] = 0 \\
\Rightarrow\ & \frac{1}{N} \sum_{i \in \mathcal{B}(s,a)} Q(s,a) - y(i) = 0 \\
\Rightarrow\ & Q(s,a) - \frac{2c}{N} \sum_{i \in \mathcal{B}(s,a)} y(i) = 0 \\
\Rightarrow\ & Q(s,a) = \frac{1}{N} \sum_{i \in \mathcal{B}(s,a)} y(i).
\end{aligned}
\tag{14}
$$

∎

**Observation 3** *(L1 Loss) Let $\mathcal{B}(s,a) \subset \mathcal{B}$ be the subset of transitions containing $(s,a)$ and $\delta(i) = Q(i) - y(i)$. If $\nabla_Q \mathbb{E}_{i \sim \mathcal{B}}[|\delta(i)|] = 0$ then $Q(s,a) = \text{median}_{i \in \mathcal{B}(s,a)} y(i)$.*

*Proof.*

$$
\begin{aligned}
& \mathbb{E}_{i \sim \mathcal{B}(s,a)}[\nabla_Q |\delta(i)|] = 0 \\
\Rightarrow\ & \mathbb{E}_{i \sim \mathcal{B}(s,a)}[\text{sign}(\delta(i))] = 0 \\
\Rightarrow\ & \sum_{i \in \mathcal{B}(s,a)} \mathbb{1}\{Q(s,a) \le y(i)\} = \sum_{i \in \mathcal{B}(s,a)} \mathbb{1}\{Q(s,a) \ge y(i)\} \\
\Rightarrow\ & Q(s,a) = \text{median}_{i \in \mathcal{B}(s,a)} y(i).
\end{aligned}
\tag{15}
$$

∎

### A.4 PAL Derivation

**Observation 4** *LAP and PAL have the same expected gradient.*

*Proof.* From Corollary 1 we have:

$$
\begin{aligned}
\mathcal{L}_{\text{PAL}}(\delta(i)) &= \frac{1}{\lambda} |pr(i)|_\times \mathcal{L}_{\text{Huber}}(\delta(i)) \\
&= \frac{1}{\lambda} |\max(|\delta(i)|^\alpha, 1)|_\times \mathcal{L}_{\text{Huber}}(\delta(i)) \\
&= \frac{1}{\lambda} |\max(|\delta(i)|^\alpha, 1)|_\times \begin{cases} 0.5\delta(i)^2 & \text{if } |\delta(i)| \le 1, \\ |\delta(i)| & \text{otherwise,} \end{cases} \\
&= \frac{1}{\lambda} \begin{cases} 0.5\delta(i)^2 & \text{if } |\delta(i)| \le 1, \\ \frac{|\delta(i)|^{1+\alpha}}{1+\alpha} & \text{otherwise,} \end{cases}
\end{aligned}
\tag{16}
$$

where

$$
\lambda = \frac{\sum_j pr(j)}{N} = \frac{\sum_j \max(|\delta(j)|^\alpha, 1)}{N}.
\tag{17}
$$

Then by Corollary 1, LAP and PAL have the same expected gradient.

∎

# B Computational Complexity Results

A bottleneck in the usage of prioritized experience replay (PER) [1] is the computational cost induced by non-uniform sampling. Most implementations of PER use a sum-tree to keep the sampling cost to $O(\log{(N)})$, where $N$ is the number of elements in the buffer. However, we found common implementations to have inefficient aspects, mainly unnecessary for-loops. Since a mini-batch is sampled every time step, any inefficiency can add significant costs to the run time of the algorithm. While our implementation has no algorithmic differences and still relies on a sum-tree, we found it significantly outperformed previous implementations.

Figure 1: The average run time increase of different implementations of PER in minutes, over 1 million time steps and averaged over 3 trials. Time increase of SAC is provided to give a better understanding of the significance. Our implementation only adds a cost of 11 minutes, per million time steps, while the OpenAI baselines implementation adds over 3 hours.

We compare the run time of our implementation of PER with two standard implementations, OpenAI baselines [2] and Dopamine [3]. To keep things fair, all components of the experience replay buffer and algorithm are fixed across all comparisons, and only the sampling of the indices of stored transitions and the computation of the importance sampling weights is replaced. Both implementations are taken from the master branch in early February 2020[1]. Each implementation of PER is combined with TD3. The OpenAI baselines implementation uses an additional sum-tree to compute the minimum over the entire replay buffer to compute the importance sampling weights. However, to keep the computational costs comparable, we remove this additional sum-tree and use a per-batch minimum, similar to Dopamine and our own implementation. Additionally, we compare against TD3 with a uniform experience replay as well as SAC [4]. All time-based experiments are run on a single GeForce GTX 1080 GPU and a Intel Core i7-6700K CPU. Our results are presented in Figure 1 and Table 1.

Table 1: Average run time of different implementations of PER, and their percentage increase over TD3 with a uniform buffer. Values are computed over 1 million time steps and averaged over 3 trials. Run time of TD3 and SAC with uniform buffers are also provided to give a better understanding of the scale. $\pm$ captures a 95% confidence interval over the run time. Fastest run time implementation of PER is bolded.

|  | TD3 + Uniform | TD3 + Ours | TD3 + Dopamine | TD3 + Baselines | SAC |
|---|---|---|---|---|---|
| Run Time (mins) | $81.62 \pm 1.14$ | $\mathbf{92.78 \pm 0.93}$ | $143.02 \pm 1.75$ | $268.80 \pm 7.39$ | $139.70 \pm 2.22$ |
| Time Increase (%) | +0.00% | **+13.68%** | +75.24% | +229.35% | +71.17% |

We find our implementation of PER greatly outperforms the other standard implementations in terms of run time. This means PER can be added to most methods without significant computational costs if implemented efficiently. Additionally, we find that TD3 with PER can be run faster than a comparable and commonly used method, SAC. Our implementation of PER adds less than 50 lines of code to the standard experience replay buffer code. We hope the additional efficiency will enable further research in non-uniform sampling methods.

# C Additional Experiments

In this section we perform additional experiments and visualizations, covering ablation studies, additional baselines and display the learning curves for the Atari results.

## C.1 Ablation Study

To better understand the contributions of each component in PAL, we perform an ablation study. We aim to understand the importance of the scaling factor $\frac{1}{\lambda} = \frac{N}{\sum_j \max(|\delta(j)|^{\alpha}, 1)}$ as well as the differences between the proposed loss and the comparable Huber loss [5], by considering all possible combinations. Notably, when $\alpha = 0$, PAL equals the Huber loss with the scaling factor. As discussed in the Experimental Details, Appendix D, PAL uses $\alpha = 0.4$. The results are reported in Figure 2 and Table 2.

We find the complete loss of PAL achieves the highest performance, and PAL without the $\frac{1}{\lambda}$ scale factor to be the second highest. Interestingly, while TD3 with the Huber loss performs poorly, scaling by $\frac{1}{\lambda}$ adds fairly significant gains in performance.

Figure 2: Learning curves for the ablation study on the suite of OpenAI gym continuous control tasks in MuJoCo. Curves are averaged over 10 trials, where the shaded area represents a 95% confidence interval over the trials.

Table 2: Average performance over the last 10 evaluations and 10 trials. $\pm$ captures a 95% confidence interval. Scores are bold if the confidence interval intersects with the confidence interval of the highest performance, except for HalfCheetah and Walker2d where all scores satisfy this condition.

|  | TD3 | TD3 + Huber | TD3 + Huber + $\frac{1}{\lambda}$ | TD3 + PAL - $\frac{1}{\lambda}$ | TD3 + PAL |
|---|---|---|---|---|---|
| HalfCheetah | $13570.9 \pm 794.2$ | $14820.5 \pm 785.5$ | $13772.2 \pm 685.7$ | $14404 \pm 642.2$ | $15012.2 \pm 885.4$ |
| Hopper | $\mathbf{3393.2 \pm 381.9}$ | $2125.7 \pm 596.9$ | $\mathbf{3442.2 \pm 319.4}$ | $\mathbf{3135.8 \pm 479.5}$ | $\mathbf{3129.1 \pm 473.5}$ |
| Walker2d | $4692.4 \pm 423.6$ | $4311.3 \pm 1219.2$ | $4707.3 \pm 435.3$ | $5313.7 \pm 368.2$ | $5218.7 \pm 422.6$ |
| Ant | $\mathbf{6469.9 \pm 200.3}$ | $4952.6 \pm 1204.2$ | $\mathbf{6499.2 \pm 162.7}$ | $\mathbf{6322.7 \pm 564.3}$ | $\mathbf{6476.2 \pm 640.2}$ |
| Humanoid | $6437.5 \pm 349.3$ | $5039.1 \pm 1631.5$ | $6163.2 \pm 331.7$ | $\mathbf{7493.4 \pm 645.3}$ | $\mathbf{8265.9 \pm 519.0}$ |

## C.2 Additional Baselines

To better compare our algorithm against other recent adjustments to replay buffers and non-uniform sampling, we compare LAP and PAL against the Emphasizing Recent Experience (ERE) replay buffer [6] combined with TD3. To implement ERE we modify our training procedure slightly, such that $X$ training iterations are the applied at the end of each episode, where $X$ is the length of the episode. ERE works by limiting the transitions sampled to only the $N$ most recent transitions, where $N$ changes over the $X$ training iterations. Additionally, we add a baseline of the simplest version of LAP which uses the L1 loss, rather than the Huber loss and samples transitions with priority $pr(i) = |\delta(i)|^{\alpha}$, denoted TD3+L1+$\alpha$. Results are reported in Figure 3 and Table 3.

We find that LAP and PAL with TD3 outperform ERE. The addition of ERE outperforms vanilla TD3 and improves early learning performance in several tasks. We remark that the addition of ERE does not directly conflict with LAP and PAL. PAL can be directly combined with ERE with no other modifications. LAP can be combined by only performing the non-uniform sampling on the corresponding subset of transitions determined by ERE. We leave these combinations to future work. We also find that LAP and PAL outperform the simplest version of LAP with the L1 loss. This demonstrates the importance of the Huber loss in LAP.

Figure 3: Learning curves for additional baselines on the suite of OpenAI gym continuous control tasks in MuJoCo. Curves are averaged over 10 trials, where the shaded area represents a 95% confidence interval over the trials.

Table 3: Average performance over the last 10 evaluations and 10 trials. $\pm$ captures a 95% confidence interval. Scores are bold if the confidence interval intersects with the confidence interval of the highest performance, except for HalfCheetah, Hopper and Walker2d where all scores satisfy this condition.

|  | TD3 | TD3 + ERE | TD3 + L1 + $\alpha$ | TD3 + LAP | TD3 + PAL |
|---|---|---|---|---|---|
| HalfCheetah | $13570.9 \pm 794.2$ | $14863.6 \pm 602.0$ | $14885.4 \pm 402.6$ | $14836.5 \pm 532.2$ | $15012.2 \pm 885.4$ |
| Hopper | $3393.2 \pm 381.9$ | $3541.7 \pm 253.3$ | $3208.9 \pm 475.1$ | $3246.9 \pm 463.4$ | $3129.1 \pm 473.5$ |
| Walker2d | $4692.4 \pm 423.6$ | $5205.5 \pm 407.0$ | $5153.7 \pm 601.4$ | $5230.5 \pm 368.2$ | $5218.7 \pm 422.6$ |
| Ant | $6469.9 \pm 200.3$ | $6285.9 \pm 271.7$ | $6021.1 \pm 656.9$ | $\mathbf{6912.6 \pm 234.4}$ | $\mathbf{6476.2 \pm 640.2}$ |
| Humanoid | $6437.5 \pm 349.3$ | $6889.9 \pm 627.4$ | $6185.3 \pm 207.0$ | $\mathbf{7855.6 \pm 705.9}$ | $\mathbf{8265.9 \pm 519.0}$ |

## C.3 Full Atari Results

We display the full learning curves from the Atari experiments in Figure 4.

Figure 4: Complete learning curves for a set of Atari games. Curves are smoothed uniformly with a sliding window of 10 for visual clarity.

# D   Experimental Details

All networks are trained with PyTorch (version 1.2.0) [7], using PyTorch defaults for all unmentioned hyper-parameters.

## D.1   MuJoCo Experimental Details

**Environment.** Our agents are evaluated in MuJoCo (mujoco-py version 2.0.2.9) [8] via OpenAI gym (version 0.15.4) [9] interface, using the v3 environments. The environment, state space, action space, and reward function are not modified or pre-processed in any way, for easy reproducibility and fair comparison with previous results. Each environment runs for a maximum of 1000 time steps or until some termination condition, and has a multi-dimensional action space with values in the range of $(-1, 1)$, except for Humanoid which uses a range of $(-0.4, 0.4)$.

**Architecture.** Both TD3 [10] and SAC [4] are actor-critic methods which use two Q-networks and a single actor network. All networks have two hidden layers of size 256, with ReLU activation functions after each hidden layer. The critic networks take state-action pairs $(s, a)$ as input, and output a scalar value $Q$ following a final linear layer. The actor network takes state $s$ as input and outputs a multi-dimensional action $a$ following a linear layer with a tanh activation function, multiplied by the scale of the action space. For clarity, a network definition is provided in Figure 5.

```
(input dimension, 256)
ReLU
(256, 256)
RelU
(256, output dimension)
```

Figure 5: Network architecture. Actor networks are followed by a `tanh · max action size`

**Network Hyper-parameters.** Networks are trained with the Adam optimizer [11], with a learning rate of $3e - 4$ and mini-batch size of $256$. The target networks in both TD3 and SAC are updated with polyak averaging with $\nu = 0.005$ after each learning update, such that $\theta' \leftarrow (1 - \nu)\theta' + \nu\theta$, as described by Lillicrap et al. [12].

**Terminal Transitions.** The learning target uses a discount factor of $\gamma = 0.99$ for non-terminal transitions and $\gamma = 0$ for terminal transitions, where a transition is considered terminal only if the environment ends due to a termination condition and not due to reaching a time-limit.

**LAP, PAL, and PER.** For LAP and PAL we use $\alpha = 0.4$. For PER we use $\alpha = 0.6$, $\beta = 0.4$ as described by Schaul et al. [1] and $\epsilon = 1e - 10$ from Castro et al. [3]. Since there are two TD errors defined by $\delta_1 = Q_\theta^1 - y$ and $\delta_2 = Q_\theta^2 - y$, where $y = r + \gamma \min(Q_{\theta'}^1(s', a'), Q_{\theta'}^2(s', a'))$ [10], the priority uses the maximum over $|\delta_1|$ and $|\delta_2|$, which we found to give the strongest performance. As done by PER, new samples are given a priority equal to the maximum priority recorded at any point during learning.

**TD3 and SAC.** For TD3 we use the default policy noise of $\mathcal{N}(0, \sigma_N^2)$, clipped to $(-0.5, 0.5)$, where $\sigma_N = 0.2$. Both values are scaled by the range of the action space. For SAC we use the learned entropy variant [13], where entropy is trained to a target of $-$`action dimensions` as described the author, with an Adam optimizer with learning $3e - 4$, matching the other networks. Following the author's implementation, we clip the log standard deviation to $(-20, 2)$, and add $\epsilon = 1e - 6$ to avoid numerical instability in the logarithm operation. We consider this a hyper-parameter as $\log(\epsilon)$ defines the minimum value subtracted from log-likelihood calculation of the tanh normal distribution:

$$\log \pi(a|s) = \sum_i -0.5 \frac{(u_i - \mu_i)^2}{\sigma_i^2} - \log \sigma_i - 0.5 \log(2\pi) - \log(1 - a_i^2 + \epsilon), \qquad (18)$$

where $i$ is each action dimension, $u$ is pre-activation value of the action $a$ and $\mu$ and $\sigma$ define the Normal distribution output by the actor network. The log-likelihood calculation assumes the action is scaled within $(-1, 1)$. As standard practice, the agents are trained for one update after every time step.

**Exploration.** To fill the buffer, for the first 25k time steps the agent is not trained, and actions are selected randomly with uniform probability. Afterwards, exploration occurs in TD3 by adding

Gaussian noise $\mathcal{N}(0, \sigma_E^2 \cdot \texttt{max action size})$, where $\sigma_E = 0.1$ scaled by the range of the action space. No exploration noise is added to SAC, as it uses a stochastic policy.

For clarity, all hyper-parameters are presented in Table 4.

Table 4: Continuous control hyper-parameters.

| Hyper-parameter | Value |
|---|---|
| Optimizer | Adam |
| Learning rate | $3e - 4$ |
| Mini-batch size | 256 |
| Discount factor $\gamma$ | 0.99 |
| Target update rate | 0.005 |
| Initial random policy steps | 25k |
| TD3 Exploration Policy $\sigma_E$ | 0.1 |
| TD3 Policy Noise $\sigma_N$ | 0.2 |
| TD3 Policy Noise Clipping | $(-0.5, 0.5)$ |
| SAC Entropy Target | $-\texttt{action dimensions}$ |
| SAC Log Standard Deviation Clipping | $(-20, 2)$ |
| SAC $\epsilon$ | $1e - 6$ |
| PER priority exponent $\alpha$ | 0.6 |
| PER importance sampling exponent $\beta$ | 0.4 |
| PER added priority $\epsilon$ | $1\,e - 10$ |
| LAP & PAL exponent $\alpha$ | 0.4 |

**Hyper-parameter Optimization.** No hyper-parameter optimization was performed on TD3 or SAC. For LAP and PAL we tested $\alpha = \{0.2, 0.4, 1\}$ on the HalfCheetah task using seeds than the final results reported. Note 0.4 comes from $\alpha - \alpha\beta = 0.36$ when using the hyper-parameters defined by Schaul et al. [1]. Additionally we tested using defining the priority over $|\delta_1|$, $0.5(|\delta_1| + |\delta_2|)$ and $\max(|\delta_1|, |\delta_2|)$ and found the maximum to work the best for both LAP and PER.

**Evaluation.** Evaluations occur every 5000 time steps, where an evaluation is the average reward over 10 episodes, following the deterministic policy from TD3, without exploration noise, or by taking the deterministic mean action from SAC. To reduce variance induced by varying seeds [14], we use a new environment with a fixed seed (the training seed + a constant) for each evaluation, so each evaluation uses the same set of initial start states.

**Visualization.** Performance is displayed in learning curves, presented as the average over 10 trials, where a shaded region is added, representing a $95\%$ confidence interval over the trials. The curves are smoothed uniformly over a sliding window of 5 evaluations for visual clarity.

## D.2    Atari Experimental Details

**Environment.** For the Atari games we interface through OpenAI gym (version 0.15.4) [9] using NoFrameSkip-v0 environments with sticky actions with $p = 0.25$ [15]. Our pre-processing steps follow standard procedures as defined by Castro et al. [3] and Machado et al. [15]. Exact pre-processing steps can be found in [16] which our code is closely based on.

**Architecture.** We use the standard architecture used by DQN-style algorithms. This is a 3-layer CNN followed by a fully connected network with a single hidden layer. The input to this network is a $(4, 84, 84)$ tensor, corresponding to the previous 4 states of $84 \times 84$ pixels. The first convolutional layer has $32$ $8 \times 8$ filers with stride 4, the second convolutional layer has $32$ $4 \times 4$ filers with stride 2, the final convolutional layer has $64$ $2 \times 2$ filers with stride 1. The output is flattened into a 3136 vector and passed to the fully connected network with a single hidden layer of $512$. The final output dimension is the number of actions. All layers, except for the final layer are followed by a ReLU activation function.

**Network Hyper-parameters.** Networks are trained with the RMSProp optimizer, with a learning rate of $6.25e - 5$, no momentum, centered, $\alpha = 0.95$, and $\epsilon = 1e-5$. We use a mini-batch size of

32. The target network is updated every 8k training iterations. A training update occurs every 4 time steps (i.e. 4 interactions with the environment).

**Terminal Transitions.** The learning target uses a discount factor of $\gamma = 0.99$ for non-terminal transitions and $\gamma = 0$ for terminal transitions, where a transition is considered terminal only if the environment ends due to a termination condition and not due to reaching a time-limit.

**LAP, PAL, and PER.** For LAP and PAL we use $\alpha = 0.6$. For PER we use $\alpha = 0.6$, $\beta = 0.4$ as described by Schaul et al. [1] and $\epsilon = 1e - 10$ from Castro et al. [3]. As done by PER, new samples are given a priority equal to the maximum priority recorded at any point during learning. For LAP, due to the magnitude of training errors being lower, rather than use a Huber loss with $\kappa = 1$, we use $\kappa = 0.01$. This gives a loss function of:

$$\mathcal{L}_{\text{Huber}}^{\kappa}(\delta(i)) = \begin{cases} 0.5\delta(i)^2 & \text{if } |\delta(i)| \leq \kappa, \\ \kappa(|\delta(i)| - 0.5\kappa) & \text{otherwise.} \end{cases} \tag{19}$$

To maintain the unbiased nature of LAP, the priority clipping needs to be shifted according to the choice of $\kappa$, generalizing the priority function to be:

$$p(i) = \frac{\max(|\delta(i)|^{\alpha}, \kappa^{\alpha})}{\sum_j \max(|\delta(j)|^{\alpha}, \kappa^{\alpha})}. \tag{20}$$

Note the case where $\kappa = 1$, the algorithm is unchanged. PAL also needs to be generalized accordingly:

$$\mathcal{L}_{\text{PAL}}(\delta(i)) = \frac{1}{\lambda} \begin{cases} 0.5\kappa^{\alpha}\delta(i)^2 & \text{if } |\delta(i)| \leq \kappa, \\ \frac{\kappa|\delta(i)|^{1+\alpha}}{1+\alpha} & \text{otherwise,} \end{cases} \qquad \lambda = \frac{\sum_j \max(|\delta(j)|^{\alpha}, \kappa^{\alpha})}{N}. \tag{21}$$

**Exploration.** To fill the buffer, for the first 20k time steps the agent is not trained, and actions are selected randomly with uniform probability. Afterwards, exploration occurs according to the $\epsilon$-greedy policy where $\epsilon$ begins at 1 and is decayed to 0.01 over the next 250k training iterations, corresponding to 1 million time steps.

For clarity, all hyper-parameters are presented in Table 5.

Table 5: Atari Hyper-parameters.

| Hyper-parameter | Value |
|---|---|
| Optimizer | RMSProp |
| RMSProp momentum | 0 |
| RMSProp centered | True |
| RMSProp $\alpha$ | 0.95 |
| RMSProp $\epsilon$ | $1e - 5$ |
| Learning rate | $6.25e - 5$ |
| Mini-batch size | 32 |
| Discount factor $\gamma$ | 0.99 |
| Target update rate | 8k training iterations |
| Initial random policy steps | 20k |
| Initial $\epsilon$ | 1 |
| End $\epsilon$ | 0.01 |
| $\epsilon$ decay period | 250k training iterations |
| Evaluation $\epsilon$ | 0.001 |
| Training frequency | 4 time steps |
| PER priority exponent $\alpha$ | 0.6 |
| PER importance sampling exponent $\beta$ | 0.4 |
| PER added priority $\epsilon$ | $1\,e - 10$ |
| LAP & PAL exponent $\alpha$ | 0.6 |

**Hyper-parameter Optimization.** No hyper-parameter optimization was performed on Double DQN. For LAP and PAL we tested $\alpha = \{0.4, 0.6\}$ on the Asterix game, which was excluded from the final results.

**Evaluation.** Evaluations occur every 50k time steps, where an evaluation is the average reward over 10 episodes, following the $\epsilon$-greedy policy where $\epsilon = 1e-3$. To reduce variance induced by varying seeds [14], we use a new environment with a fixed seed (the training seed + a constant) for each evaluation, so each evaluation uses the same set of initial start states. Final scores reported in Figure 2 and Table 2 average over the final 10 evaluations. For Table 2 percentage improvement of $x$ over $y$ is calculated by $\frac{x-y}{|y|}$.

## Footnotes

[1]OpenAI baselines `https://github.com/openai/baselines`, commit: ea25b9e8b234e6ee1bca43083f8f3cf974143998, Dopamine `https://github.com/google/dopamine`, commit: e7d780d7c80954b7c396d984325002d60557f7d1.