[Reviews · NeurIPS 2020]

Review 1

Summary and Contributions: In this paper they argue that there are two different contributions to PER. On the one hand there is the change in the direction of the gradient based on the prioritization schemes and on the other hand there is the non-uniformity of the updates themselves (which relates to the rate of signal propagation throughout the TD learning). They show a connection to changing the loss itself and leaving the sampling scheme uniform and perform multiple experiments comparing the various methods.

Strengths: The paper is well written and is a novel approach. Specifically, the ability to map a priorized sampling scheme to an equivalent (in expectation) gradient with respect to a uniform sampling scheme may drive many future research directions and innovations.

Weaknesses: While in expectation the gradient of PAL and LAP are identical, the behavior in complex domains is not. Investigating this question is very important and is left as an open hole in the work. One explanation that could hold is that although the gradient is identical in expectation, the prioritization scheme causes the signal to propagate faster. In this case, the difference between LAP and PAL may be tied to the periodicy of the MDP. MuJoCo domains are very simple and repetitive, thus random uniform sampling is probably sufficient for signal propagation and in the ATARI domains the prioritization provides a large benefit.

Correctness: Yes.

Clarity: Very.

Relation to Prior Work: Yes.

Reproducibility: Yes

Additional Feedback: I enjoyed reading this paper. The idea is very interesting and novel. Very simple to grasp, very simple to implement. I believe this work can have a real impact (in terms of adoption) on the community. ** post rebuttal ** I really liked this paper. I think that it can be improved by looking into the effects of each method based on the periodicity of the MDP.


Review 2

Summary and Contributions: This paper analyzes the loss function induced by prioritized experience replay (PER), and finds that it is possible to remove the importance sampling ratio in the loss used with PER by instead using a different loss altogether. The theoretical analysis provides a more general analysis of what losses can be used when sampling from a distribution different than uniformly sampling the replay buffer. This analysis leads to the paper presenting a new prioritization scheme, which they call LAP, and a new loss, which is referred to as PAL. Experiments compare these two new schemes to methods that use a replay buffer in both the continuous control and atari benchmarks.

Strengths: * A deeper analysis of methods that have been found to be empirically useful is of immense value to the community. In this case, the perspective of considering the loss function that PER induces and then attempting to remove the requirement for an importance sampling ratio are commendable. * Further breaking down the benefit of PER as variance reduction due to non-uniform sampling and changing the expected gradient due to the induced loss function is a useful insight. * Encouraging Experimental results

Weaknesses: Section 5 was harder to follow than the rest of the paper. If I understand correctly, the LAP procedure aims to use the Huber loss after sampling non-uniformly rather than using the L1 loss. In order to make this loss unbiased the LAP procedure suggests a new prioritized sampling procedure. PAL then proposes a loss function to be used if we want to to sample from the buffer uniformly but would like to have the same expected gradient as LAP. This idea could be put forth a little more clearly.

Correctness: There do not seem to be any glaring issues in the correctness of the theoretical results. **After Rebuttal** I am satisfied with the empirical methodology for the most part. In the experiments on continuous control, the separation in the performance of SAC and the proposed techniques happens just barely at the point where they are evaluated. Perhaps a better method to compare the two techniques is to look at the area under the learning curve or evaluation at various points in learning. Authors did not respond to this suggested metric.

Clarity: I have some issue with the clarity of the paper. Various terms are used without previous introduction. Examples are the horizon T and target network parameters \theta' in Section 3. Another ambiguous statement is the summation when computing p(i) just before Equation (6). The summation in the denominator is not clear about what range it is summed over. I assume it is batch D, but would be helpful to be clear. Another clarification is whether \delta(j) in the denominator should also be an absolute value. **After Rebuttal** Authors have agreed to look into above points.

Relation to Prior Work: Relation to previous work is discussed sufficiently

Reproducibility: Yes

Additional Feedback: The paper suffers in its readability and clarity. See comments above. Additional discussion with the other reviewers and AC led to the following comments: 1. Corollary 1 seems not quite correct. They're applying the fundamental theorem of calculus so that a gradient and an integral cancel out. But this requires that the function inside the integral is continuous. While this is quite mild, the function inside is itself a gradient! So if we had a ReLU network, the gradient of the loss would be highly discontinuous. 2. Corollary 1 is also stated in a weird way. In Theorem 1, the conclusion is that L_1 and \lambda L_2 have the same expected gradients. But because they have 1/\lambda in the expression in Corollary 1, you actually find that L_1 and L_2 themselves have the same expected gradients. So formally Theorem 1 is not satisfied by this construction. They should either remove the 1/\lambda or change Theorem 1.


Review 3

Summary and Contributions: This papers examines prioritized experience replay, where transitions are sampled in a non-uniform manner to constitute the optimization objective, and presents a new uniformly sampled loss function that yields the same gradients in expectation. The paper provides a theoretical derivation and also shows promising empirical results that back up the claims. Update: Thanks to the authors for their clarifications!

Strengths: * Novel theoretical contribution of demonstrating why Prioritized Experience Replay works * A new loss function (loss-adjusted PER) and sampling algorithm (PAL and LAP) for updating RL agents * Promising empirical results

Weaknesses: * No major weaknesses. If I were to point out something, it would be that the paper could perhaps use some empirical analyses (beyond learning curves on different environments), probing what conditions or assumptions benefit PAL and LAP over traditional PER.

Correctness: * Claims seem largely correct to me

Clarity: * The paper is very well written! I appreciate the way the connection between sampling and loss functions was established in Section 4 before deriving the new schemes in section 5.

Relation to Prior Work: * yes, related work is well represented.

Reproducibility: No

Additional Feedback: * It would help to provide details on exact models, training schemes, etc. used for reproducibility. Releasing the code would be even better. * Line 6 (abstract): The sentence leading with 'surprisingly' put me off a bit in the abstract without further information that comes later in the paper -- if the theory is showing they are equivalent, why should the empirical results working out be surprising?


Review 4

Summary and Contributions: This paper argues that original PER has bias and it requires importance sampling. In order to solve this problem, the authors derive an unbiased loss condition for a given priority scheme. From this derivation, the authors suggest an unbiased prioritization method without importance sampling by using the Huber loss (LAP), and loss with uniform sampling, which is equivalent to LAP (PAL).

Strengths: The paper suggests a prioritization method with unbiased loss for given priority scheme and converts it to equivalent loss with uniform sampling from the buffer. Furthermore, LAP considers a minimum variance criterion when TD error is large (L1 loss). By applying LAP to TD3 and DDQN, the performance is enhanced notably. Also, PAL uses equivalent loss with uniform sampling, which has the same expectation of loss gradient. Thus, it does not require complex prioritization scheme maintaining the advantage of prioritization.

Weaknesses: The paper argues that PAL can avoid prioritization scheme by using equivalent loss, but the performance of PAL is much lower than that of LAP. As explained in the paper, LAP considers the minimum variance criterion, but PAL cannot reflect the criterion. Thus, PAL cannot completely replace LAP and one still has to consider prioritized sampling from the buffer to have advantage of prioritization. LAP enhances the performance of baseline algorithms based on unbiased estimation and variance reduction, but the enhancement is limited for specific tasks. For example, the performance of other tasks except Humanoid in continuous domain does not increase much. That is probably because TD3 already works well in most tasks, so it seems to be necessary to do additional experiments.

Correctness: The suggested prioritization method is correctly unbiased without importance sampling and the empirical methodology is also correct.

Clarity: The overall flow of the paper is well written. The proof of theorem is easy to understand and several observations and corollary help understanding.

Relation to Prior Work: It gives clear description for the suggested prioritization method and the difference from the existing PER method.

Reproducibility: Yes

Additional Feedback: ====== After author response ====== The authors wrote the feedback well, and I understood and accepted it.

[Author Response · NeurIPS 2020]

We thank the reviewers for their time, comments and feedback. The reviewers unanimously appreciated the significance of our theoretical and empirical contributions. As a result, this rebuttal will mainly focus on minor comments and questions.

**(R1, R4) – Analysis on LAP vs. PAL.** A common question was with respect to the understanding the performance difference between LAP and PAL. Our theoretical analysis shows that prioritization reduces the variance of the gradient. Given LAP and PAL have the same expected gradient, our hypothesis is that LAP offers performance gains over PAL when the variance reduction matters. As suggested by R1, in an environment like MuJoCo with dense rewards, this may have limited impact. However, in Atari which has sparse rewards, and therefore, higher variance in the gradient, prioritization samples the sparse rewards more frequently and propagates the signal faster. In the camera-ready version we will attempt to validate this hypothesis in some simple toy domains, where the properties of the environment as well as the gradient of the loss function can be more readily analyzed.

**(R2) – The authors also claim that the suggested methods are faster in execution, which is a useful side-effect if verified.** Run time improvements appear in two ways. When using PAL (over LAP or PER), we save the cost of building and sampling from the priority structure. For LAP/PER, we don't find a meaningful run time difference, however we did find our implementation of PER outperformed existing, commonly used implementations. This result can be found in Appendix B. Notably the run time TD3 + LAP is less than using vanilla SAC.

**(R2) – How was the number of time steps for the experiment decided upon?** The choice of a horizon of 3 million is a common choice in the literature. SAC [1, 2] and ERE [3] use 3 million in most environments, and OAC (from NeurIPS 2019) [4] uses 2.5 million. We fixed 3 million across all environments as we felt this was the most transparent choice.

**(R2) – Various terms are used without previous introduction. [...]** Thank you for pointing these out. We will add definitions in the camera-ready version.

**(R2) – Another ambiguous statement is the summation when computing p(i) just before Equation (6).** The summation is indeed over the batch. We will clarify this.

**(R2) – Another clarification is whether $\delta(j)$ in the denominator should also be an absolute value.** The absolute value is correct and comes from the prioritization scheme being the absolute value of the TD error.

**(R3) – Reproducibility.** We include several pages of these details in Appendix D along with code submitted in the supplementary. Please check these out. We believe it will satisfy your concerns. We absolutely believe in being as transparent and reproducible as possible.

**(R3) – The sentence leading with 'surprisingly' put me off a bit in the abstract without further information that comes later in the paper [...]** We considered this result surprising as PER has some intuitive benefits regarding signal propagation, but we agree the language could be clarified here. Thank you.

**(R4) – Broader Impact.** We will expand our discussion in the broader impact. Thank you for your comments.

# References

[1] Tuomas Haarnoja, Aurick Zhou, Pieter Abbeel, and Sergey Levine. Soft actor-critic: Off-policy maximum entropy deep reinforcement learning with a stochastic actor. In *International Conference on Machine Learning*, volume 80, pages 1861–1870. PMLR, 2018.

[2] Tuomas Haarnoja, Aurick Zhou, Kristian Hartikainen, George Tucker, Sehoon Ha, Jie Tan, Vikash Kumar, Henry Zhu, Abhishek Gupta, Pieter Abbeel, et al. Soft actor-critic algorithms and applications. *arXiv preprint arXiv:1812.05905*, 2018.

[3] Che Wang, Yanqiu Wu, Quan Vuong, and Keith Ross. Towards simplicity in deep reinforcement learning: Streamlined off-policy learning. *arXiv preprint arXiv:1910.02208*, 2019.

[4] Kamil Ciosek, Quan Vuong, Robert Loftin, and Katja Hofmann. Better exploration with optimistic actor critic. In *Advances in Neural Information Processing Systems*, pages 1787–1798, 2019.


[Meta-Review · NeurIPS 2020]

The paper provides some theoretical treatment of prioritized experience replay, and shows how the weighted sampling scheme can be viewed as minimizing a different loss function under the uniform sampling scheme. The main insight here is that since the weights are derived from the loss itself, there is some cancellation that changes the original loss function. The insights are used to derive two new algorithms which perform reasonably well in experiments. The paper is interesting, uses new theoretical insights to derive algorithms with competitive performance. As such, we recommend acceptance.